# Biomarkers in the prediction of contrast media induced nephropathy – the BITCOIN study

Felix S. Seibert[1]*, Anja Heringhaus[1], Nikolaos Pagonas[2], Henrik Rudolf[3], Benjamin Rohn[1], Frederic Bauer[1], Nina Timmesfeld[3], Hans-Joachim Trappe[4], Nina Babel[1], Timm H. Westhoff[1]

1 Department of Nephrology, University Hospital Marien Hospital Herne, Ruhr-University of Bochum, Herne, Germany, 2 Department for Angiology, Centre for Internal Medicine I, Brandenburg Medical School Theodor Fontane, Brandenburg, Germany, 3 Department of Medical Informatics, Biometry & Epidemiology, Ruhr-University of Bochum, Herne, Germany, 4 Department of Cardiology, University Hospital Marien Hospital Herne, Ruhr-University of Bochum, Herne, Germany

* felix.seibert@elisabethgruppe.de

**Data Availability Statement:** Data are available from the figshare database (DOI: 10.6084/m9. figshare.12369521).

## Abstract

### Background

Subjects with chronic kidney disease are at increased risk for contrast-induced acute kidney injury (CI-AKI). Risk stratification is traditionally based on glomerular filtration rate (GFR) and proteinuria. The present trial examines, whether tubular and inflammatory biomarkers are able to identify subjects at increased risk as well.

### Methods

We performed a prospective study in 490 patients undergoing coronary angiography. An increase of serum creatinine concentration $\geq$ 0.3 mg/dl from baseline to day 2–3 was defined as primary endpoint (CI-AKI). Urinary neutrophil gelatinase-associated lipocalin (NGAL), kidney injury molecule-1 (KIM-1), and calprotectin were assessed < 24h before coronary angiography. Prognostic accuracy was assessed by receiver operating characteristics (ROC) calculations.

### Results

30 (6.1%) patients suffered from CI-AKI (27 AKIN stage I, 3 AKIN stage II, 0 AKIN stage III). Those subjects who developed CI-AKI had 3.1 fold higher baseline urinary NGAL/creatinine ratios than those without CI-AKI (60.8 [IQR 18.7–93.1] µg/mg vs. 19.9 [IQR 12.3–38.9] µg/ mg, p = 0.001). In those subjects without clinically overt CKD (eGFR > 60 ml/min, urinary albumin creatinine ratio <30 mg/g), the NGAL/creatinine ratio was 2.6 higher in CI-AKI vs. no CI-AKI (47.8 [IQR 11.8–75.3] vs. 18.6 [IQR 11.7–36.3] µg/mg). No significant differences were obtained for KIM-1 and calprotectin (p>0.05 each). ROC analyses revealed an area under the curve (AUC) of 0.68 (95% CI 0.60–0.81) for NGAL/creatinine. An NGAL/creatinine ratio < 56.4 µg/mg has a negative predictive value of 96.5%.

**Funding:** The study was funded by the German Research Foundation (Research Unit FOR1368; receiver TW). The funders had no role in study design, data collection and analysis, decision to publish, or preparation of the manuscript.

**Competing interests:** The authors have declared that no competing interests exist.

## Conclusions

The present study is the largest investigation on the use of urinary biomarkers for CI-AKI risk stratification so far. It shows that NGAL provides prognostic information beyond the glomerular biomarkers eGFR and proteinuria.

## Introduction

Every year, approximately 75 million radiocontrast media applications take place worldwide. [1] These administrations are associated with a risk to deteriorate renal function, especially in those subjects with preexisting chronic kidney disease (CKD). Contrast-induced acute kidney injury (CI-AKI) is one of the most frequent forms of in-hospital kidney injury. Since there is no available treatment and since even a mild 0.3 mg/dl increase of serum creatinine (AKI stage I) increases in-hospital mortality by 80%, preventive measures like plasma expansion by intravenous volume application are used in subjects at increased risk. [2] Risk assessment is traditionally based on the calculation of estimated glomerular filtration rate (eGFR) and albuminuria. For two reasons it may be questioned, whether these markers are the optimal parameters to identify subjects at risk. First, eGFR calculations are imprecise in the range beyond 60 ml/min. Hence, eGFR is unable to detect mild tubular injury. Second, albuminuria is a marker of glomerular injury, not a marker of tubular injury. Contrast media, however, exert their nephrotoxicity at the tubules, both by direct cytotoxic effects and the induction of tubular hypoxia. [3]

We therefore hypothesized that urinary biomarkers of tubular injury might serve as alternative biomarkers in the risk assessment of CI-AKI. The present prospective cohort study characterizes the prognostic accuracy of neutrophil gelatinase-associated lipocalin (NGAL) as a marker of distal tubular damage, kidney injury molecule-1 (KIM-1) as a marker of proximal tubular damage, and calprotectin as an inflammatory renal biomarker for the development of CI-AKI in a population of 490 subjects undergoing coronary angiography.

## Materials and methods

### Study design and protocol

The present prospective cohort study was conducted at two German university hospitals (Ruhr-University Bochum and Charité –Universitätsmedizin Berlin, Campus Benjamin Franklin). Inclusion criterion was the indication for a coronary angiography. Exclusion criteria were acute hemodynamic shock, obstructive uropathy, urothelial carcinoma, metastatic cancer, and leukocyturia in semiquantitative dipstick examination > 1. Preexisting CKD was defined according to KDIGO criteria. [4] Subjects with an eGFR < 60 ml/min and/or an albuminuria > 30 mg/g creatinine were regarded as "clinically overt kidney disease". Coronary angiographies were performed via radial or femoral arteries. The volume of contrast media was documented. Preventive plasma expansion was performed according to physicians' clinical assessment. Physicians were blinded for the results of biomarker assessments. Blood and urine was collected 24h before contrast application and intravenous plasma expansion. In order to avoid systemic effects, we used biomarker assessments in the urine. 48 to 72h after coronary angiography a second sample was obtained. eGFR was calculated by means of the MDRD formula at both time points. AKI was defined according to the AKIN criteria. [5] We investigated the patients for subclinical renal injury by performing a postprocedural biomarker

concentration analysis (NGAL, KIM-1) parallelly to the assessment of serum creatinine. We regarded an increase of NGAL/creatinine or KIM-1/creatinine by > 100% (2 fold) as subclinical AKI. The study was approved by the local ethics committee of the Ruhr-University Bochum (registry number 4866–13) and the Charité –Universitätsmedizin Berlin (registry number EA4/117/13). The study was conducted according to the principles from the Declaration of Helsinki. All patients provided written informed consent.

## Measurement of urinary NGAL, KIM-1 and calprotectin concentrations

Urine samples (10 ml) were collected and stored frozen (-20˚C) until measurement of biomarker concentrations took place. Urinary concentration of NGAL (BPD-KIT-036, BioPorto Diagnostics), KIM-1 (ADI-900-226-0001, Enzo Life Science), and calprotectin (PhiCal® Calprotectin, K 6928, Immundiagnostik AG, Bensheim, Germany) were assessed by enzyme-linked immunosorbent assay (ELISA) according to manufacturer's protocol and previous publications. [6, 7] All the ELISA antibodies were used in previous large studies on AKI. Thus, calprotectin and KIM-1 antibodies were successfully used before in adult, pediatric and transplant populations. [6–17] The NGAL assay underwent a clinical validation and was used in diverse studies. [18] All urinary biomarker concentrations were normalized to urinary creatinine concentration.

## Statistical analysis

Data were checked for Gaussian distribution (D'Agostino Pearson). Data are presented as median and interquartile range (IQR). Comparison of continuous parameters in subjects with and without CI-AKI was performed by a Mann-Whitney U-test. Categorical parameters were compared by $\chi^2$ test. The analysis of pre- vs. postprocedural biomarkers was done via Mann-Whitney paired U-test. Receiver-operating characteristic (ROC) curves were formed in an attempt to determine the accuracy of the urinary biomarkers NGAL, KIM-1, and calprotectin in the prediction of CI-AKI, each of them adjusted for urinary creatinine. Areas under the curve (AUC) of each biomarker were compared according to DeLong. [19] Optimal cut-off values were calculated by the Youden's index. Positive predictive values and negative predictive values were calculated from sensitivity and specificity of the biomarker within the population. We performed binary logistic regression analysis for each biomarker (urinary NGAL-, KIM-1- and calprotectin/creatinine), as well as a multivariable regression model for CI-AKI involving eGFR, NGAL-, KIM-1-, and calprotectin ratios as predictors. We applied the score based prediction models from Inohara et al. and Ghani et al. to the study population, including NGAL as an additional predictor. [20, 21] P<0.05 was regarded statistically significant. All statistical analyses were performed using Prism 7 (GraphPad Software, La Jolla, CA, USA) and SPSS Statistics 25 (SPSS Inc., Chicago, IL, USA).

## Results

490 subjects (363 male, 127 female) were enrolled in the study and received a follow-up examination at 48-72h post coronary angiography. Median age was 66 (IQR 57–73). 144 (29.3%) patients suffered from pre-existing CKD, comprising 97 females and 47 males with a median age of 73 (IQR 67–78). The average amount of contrast media needed was 80 (IQR 60–120) ml. The epidemiological data are shown in Table 1. Preprocedural assessment of urinary calprotectin and NGAL was successful in the whole study population, KIM-1 failed in one sample. Follow-up urine was obtained in 472 patients. Periprocedural plasma expansion was performed in 120 subjects (24.5%).

**Table 1. Study population.**

| | Total study population (n = 490) | CI-AKI (n = 30) | No CI-AKI (n = 460) | p | Main comorbidities | |
|---|---|---|---|---|---|---|
| | | | | | Hypertension | 386 (78.8%) |
| Female | 127 (25.9%) | 9 (30%) | 108 (23.5%) | 0.386 | Diabetes | 126 (25.7%) |
| Male | 363 (74.1%) | 21 (70.0%) | 352 (76.5%) | | Coronary Heart Disease | 323 (65.9%) |
| Age (years) | 66 (57–73) | 70.5 (63–76) | 65 (56–73) | **0.013** | Medication | |
| eGFR (ml/min) | 78.6 (63.4–91.7) | 69.7 (52.9–92.0) | 78.8 (63.6–91.8) | 0.197 | ACE-I or ARB | 367 (74.9%) |
| eGFR >60 ml/min | 388 (79.2%) | 20 (66.7%) | 369 (80.2%) | 0.158 | Diuretics | 109 (22.2%) |
| eGFR 30–60 ml/min | 99 (20.2%) | 10 (33.4%) | 88 (19.1%) | | Mineralocorticoid receptor antagonist | 30 (6.1%) |
| eGFR <30 ml/min | 3 (0.6%) | 0 (0%) | 3 (0.7%) | | Betablockers | 344 (70.2%) |
| Preprocedural creatinine | 1.0 (0.8–1.1) mg/dl | 1.0 (0.8–1.3) mg/dl | 1. (0.8–1.1) mg/dl | 0.414 | Alpha 1 blockers and CCB | 131 (26.7%) |
| Postprocedural creatinine | 1.0 (0.9–1.2) mg/dl | 1.5 (1.2–1.6) mg/dl | 1.0 (0.9–1.2) mg/dl | **0.001** | | |
| AKIN stage I | 27 (5.5%) | 27 (90.0%) | - | | | |
| AKIN stage II | 3 (0.6%) | 3 (10.0%) | - | | | |
| AKIN stage III | 0 (0.0%) | 0 (0.0%) | - | | | |
| | No overt CKD (n = 346) | CI-AKI (n = 15) | No CI-AKI (n = 331) | | | |
| Female | 80 (23.1%) | 3 (20.0%) | 77 (23.3%) | 0.999 | | |
| Male | 266 (76.9%) | 12 (80.0%) | 254 (76.7%) | | | |
| Age (years) | 61 (54–70) | 70 (59–72) | 61 (54–69) | 0.130 | | |
| eGFR (ml/min) | 81.9 (75.7–94.5) | 83.4 (66.3–105.2) | 81.9 (73.9–94.4) | 0.976 | | |
| Preprocedural creatinine | 0.9 (0.80–1.0) mg/dl | 0.9 (0.8–1.2) mg/dl | 0.9 (0.8–1.0) mg/dl | 0.759 | | |
| Postprocedural creatinine | 1.0 (0.80–1.1) mg/dl | 1.3 (1.1–1.5) mg/dl | 1.0 (0.8–1.1) mg/dl | **0.001** | | |
| AKIN stage I | 14 (4.1%) | 14 (9.3%) | - | | | |
| AKIN stage II | 1 (0.3%) | 1 (0.7%) | - | | | |
| AKIN stage III | 0 (0.0%) | 0 (0.0%) | - | | | |

CKD–chronic kidney disease. CI-AKI–contrast media induced acute kidney injury. eGFR–estimated glomerular filtration rate. AKIN–acute kidney injury network.

ACE-I–angiotensin converting enzyme inhibitor. ARB—angiotensin II receptor blocker. CCB–calcium channel blockers.

CI-AKI occurred in 30 patients (6.1%). 27 (5.5%) AKIs were classified as AKIN stage I, three (0.6%) as AKIN stage II, and none as AKIN stage III (Table 1). The median urinary NGAL/creatinine ratios of patients suffering from CI-AKI (60.8 [IQR 18.7–93.1] μg/mg) were significantly higher than those without CI-AKI (19.9 [IQR 12.3–38.9] μg/mg; p = 0.001). Calprotectin/creatinine ratios showed no significant difference in the two groups (175.8 [IQR 24.1–807.1] vs. 102.2 [IQR 25.7–453.9] ng/mg; p = 0.523). Analogously, KIM-1/creatinine ratios did not differ between these groups (1249.0 [IQR 616.6–2502.0] vs. 1056.0 [IQR 700.7–1601.0] pg/mg; p = 0.145). An overview of the biomarker concentrations is presented in Table 2 and Fig 1. We also saw a postprocedural increment of all three urinary biomarker ratios (NGAL p = 0.005, KIM-1 p = 0.001, calprotectin p = 0.001). In n = 131 subjects postprocedural NGAL/creatinine ratios and in n = 92 KIM-1/creatinine ratios were increased by a factor > 2 compared to baseline values. The AUC for the prediction of these "subclinical AKIs" were as follows: Baseline NGAL/creatinine AUC 0.66 (95% CI 0.60–0.71), baseline KIM-1 0.55 (95% CI 0.48–0.61).

In the subgroup of patients, who did not have pre-existing CKD (n = 346), CI-AKI occurred in 15 patients. The severity of CI-AKI was mild (14 AKIN I, 1 case of AKIN II). Median eGFR

**Table 2. Predictive value of urinary biomarkers for contrast media induced acute kidney injury (CI-AKI).**

| | CI-AKI | No CI-AKI | p | Sensitivity | Specificity | Positive predictive value | Negative predictive value |
|---|---|---|---|---|---|---|---|
| Study population | 30 (6.1%) | 460 (93.9%) | | | | | |
| NGAL (μg/mg crea) | 60.8 (18.7–93.1) | 19.9 (12.3–38.9) | **0.001** | 82.6% | 53.3% | 16.7% | 96.5% |
| Calprotectin (ng/mg crea) | 175.8 (24.1–807.1) | 102.2 (25.7–453.9) | 0.52 | 90.0% | 23.3% | 13.7% | 94.7% |
| KIM-1 (pg/mg crea) | 1249.0 (616.6–2502.0) | 1056.0 (700.7–1601.0) | 0.15 | 82.8% | 43.3% | 12.8% | 95.7% |

Neutrophil gelatinase-associated lipocalin (NGAL), kidney injury molecule-1 (KIM-1), and calprotectin in the prediction of contrast media induced acute kidney injury (CI-AKI). All urinary biomarkers are presented after normalization for urine creatinine concentration (crea). Numeric data is presented as median with interquartile range (in brackets). P <0.05 is considered statistically significant.

of these 346 patients was 81.9 (IQR 73.7–94.5) ml/min with a median ACR of 4.2 (IQR 2.7–7.8) mg/g creatinine. NGAL/creatinine ratios were 2.6 fold higher in those subjects, who later developed CI-AKI (47.8 [IQR 11.8–75.3] vs. 18.6 [IQR 11.7–36.3] μg/mg), but failed to reach significance (AUC 0.63 [95% CI 0.46–0.79]; p = 0.102). The urinary creatinine ratios of KIM-1 (867.5 [IQR 509.7–1466] vs. 989.1 [IQR 636–1445] pg/mg; p = 0.703) and calprotectin (55.5 [IQR 9.4–634.4] vs. 90.9 [IQR 21.4–399] ng/mg; p = 0.576) were similar in both groups. The results of the biomarker analyses are presented in Fig 2. Postprocedural urinary analysis revealed an increase of all biomarkers (NGAL p = 0.015, KIM-1 p = 0.001, Calprotectin p = 0.032). 95 and 57 patients showed an increase of NGAL/creatinine and KIM-1/creatinine by 100%, respectively. The calculated AUC in ROC analysis were 0.67 (95% CI 0.61–0.74) for NGAL/creatinine and 0.54 (95% CI 0.48–0.61) for KIM-1/creatinine ratios.

The prognostic accuracy of NGAL-, KIM-1- and calprotectin/creatinine ratios were analysed by ROC analyses and displayed in Fig 3. Univariate regression analysis was significant for NGAL only (p = 0.005), yielding the highest prognostic accuracy with an AUC of 0.68 (95% CI 0.60–0.81). The AUC of KIM-1 and calprotectin were 0.58 (95% CI 0.46–0.70) and 0.54 (95% CI 0.42–0.65). The pairwise comparison of urinary NGAL- and calprotectin/creatinine AUCs showed a significant difference (p = 0.009). The AUC of the regression model plotting NGAL, KIM-1, calprotectin ratios as well as eGFR was 0.68 (95% CI 0.60–0.81; S1 Table). Table 2 presents the prognostic accuracy of each parameter using the optimal cut-off values obtained by Youden's index ($J$ = sensitivity + specificity—1). NGAL (56.4 μg/mg creatinine; $J$ = 0.360) revealed 82.6% sensitivity, 53.3% specificity, 16.7% positive predictive value (PPV), and 96.5% negative predictive value (NPV). In the subgroup of patients without CKD, the predictors NGAL-, KIM-1- and calprotectin/creatinine failed to reach individual significance in the regression analysis, the multivariable model yielded an AUC of 0.62 (95% CI 0.46–0.77; S1 Table).

Applying the two external CI-AKI prediction models from Inohara et al. and Ghani et al. to the present study group, led to an AUC of 0.68 (95% CI 0.60–0.76) and 0.57 (95% CI 0.46–0.67), respectively. There was a significant increase of the AUC in the Ghani model (0.69 [95% CI 0.58–0.80]; p = 0.045), when adding NGAL/creatinine as an additional predictor, whereas in case of the Inohara model, there was still a tendency of amelioration (0.73 [95% CI 0.63–0.82], p = 0.085).

## Discussion

The present work constitutes the largest prospective study investigating the predictive value of urinary biomarkers in the risk stratification of CI-AKI so far. Whereas NGAL and KIM-1 have been investigated in this context in smaller studies before, calprotectin was examined for the first time. [22–24] Since these biomarkers are able to detect subclinical tubular injury, it

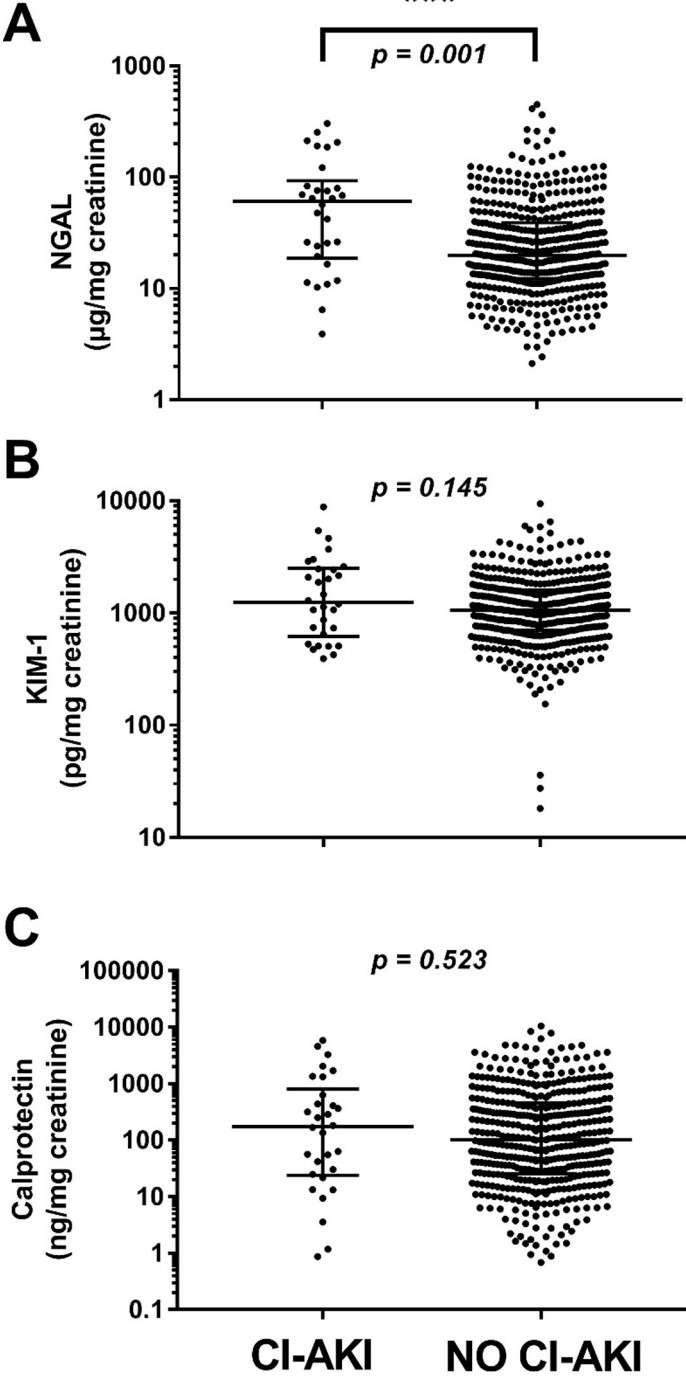

**Fig 1.** Individual urinary biomarker-creatinine ratios of subjects with and without contrast media induced acute kidney injury (CI-AKI) after coronary angiography for (A) neutrophil gelatinase-associated lipocalin (NGAL), (B) kidney injury molecule-1 (KIM-1), and (C) calprotectin. Data are presented as scatter plots (logarithmic Y-axis, medians and interquartile ranges are indicated by horizontal lines). Significant differences were ***$p < 0.001$, **$p < 0.01$ and *$p < 0.05$ by Mann-Whitney testing.

appeared reasonable that they could contribute to the risk assessment of CI-AKI, especially in those subjects without clinically overt CKD. Indeed, urinary NGAL/creatinine ratios were 3.1 times and thereby significantly higher in those subjects, who later developed CI-AKI.

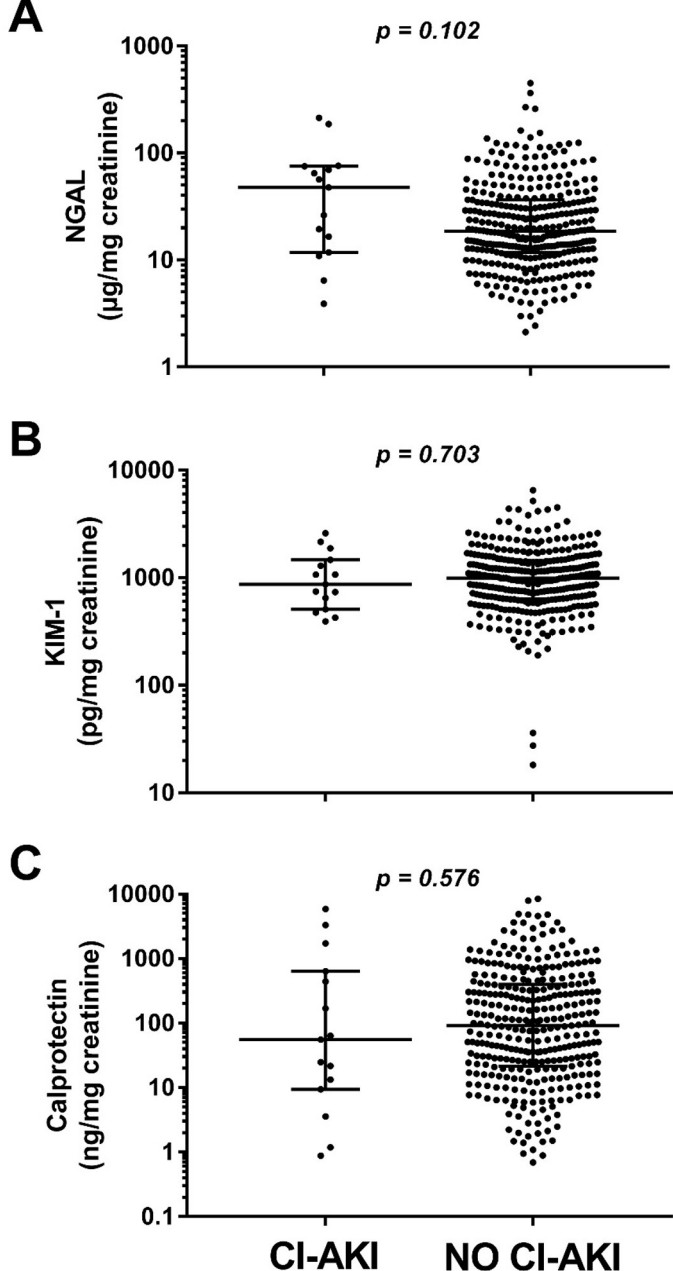

**Fig 2. Individual urinary biomarker-creatinine ratios of subjects with and without contrast media induced acute kidney injury (CI-AKI) after coronary angiography in the subgroup of patients without clinically overt kidney disease (eGFR > 60 ml, ACR > 30 mg/g creatinine; left column).** Analyses are presented for (A) neutrophil gelatinase-associated lipocalin (NGAL), (B) kidney injury molecule-1 (KIM-1), and (C) calprotectin. Data are presented as scatter plots (logarithmic Y-axis, medians and interquartile ranges are indicated by horizontal lines). Significant differences were ***p<0.001, **p<0.01 and *p<0.05 by Mann-Whitney testing.

NGAL served as a marker of distal tubular injury in the present study. It has been repeatedly described as an early diagnostic marker after contrast application, but data on its predictive value for CI-AKI is available only from a few very small trials with conflicting results. [22–24] In the present study it proved the best prognostic accuracy of the investigated urinary

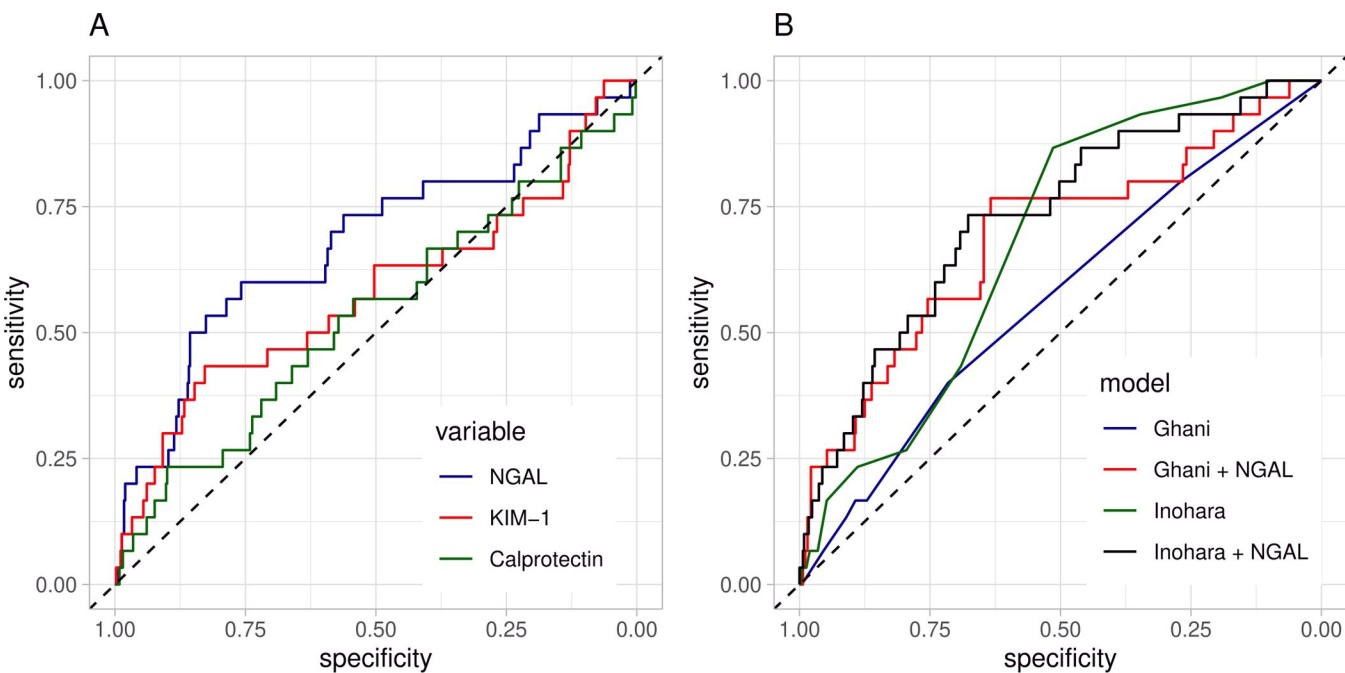

**Fig 3.** Accuracy of biomarker-creatinine ratios of (A) urinary neutrophil gelatinase-associated lipocalin (NGAL, blue), kidney injury molecule-1 (KIM-1, red) and calprotectin (green) in the prediction of contrast media induced acute kidney injury (CI-AKI) after coronary angiography in receiver operating characteristic (ROC) analysis. The predictive accuracy for CI-AKI in the present study population following the models of Ghani et al. (blue) and Inohara et al. (green) is displayed in (B). ROC curves adding NGAL/creatinine as predictor into the model are displayed in red and black, respectively. AUC–area under the curve. Diagonal scattered lines indicate prediction of CI-AKI by chance.

biomarkers with a high negative predictive value. Thus, an NGAL/creatinine concentration < 56.4 μg/mg precludes the occurrence of CI-AKI with a 96.5% probability.

KIM-1 was included in the study as a marker of proximal tubular injury. In analogy to NGAL it was demonstrated, that KIM-1 can be a useful diagnostic tool for an early detection of CI-AKI. [23] There are no data, however, on a predictive value. The present findings show that KIM-1 is not useful for risk stratification before contrast media application. Urinary calprotectin, a danger associated molecular pattern protein of the innate immune system, mirrors the inflammatory reaction after tubular injury and is thereby able to differentiate subjects with prerenal and intrinsic tubular injury. [6, 8, 9, 25] Moreover, calprotectin plays a key role in CI-AKI by activation of toll-like receptor 4. [26] In analogy to KIM-1, however, it does not predict tubular injury after contrast application in the present study. Urinary calprotectin levels do not only reflect renal inflammation but are substantially increased in leukocyturia, e. g. in urinary tract infection. In the present population, 14.7% were tested positive for leukocyturia in dip-stick examination, which might partially explain the lacking prognostic value.

Beyond the biomarker investigations, the present study shows once more that the risk of CI-AKI is substantially lower than reported in the past. Only 6.1% of the overall population and 10.4% of the CKD population fulfilled the criteria of an AKI, the majority corresponding to AKIN stage I. There was no severe CI-AKI corresponding to AKIN stage III. A decade ago, the Oxilan Registry described a CI-AKI incidence of 10.5% after radiocontrast media application. [27] In analogy with our findings, the incidences in the recent PRESERVE and AMACING trials were rather low (4.4, 4.7% and 2.6, 2.7%, respectively). [28, 29] Less toxic contrast media, less amounts of contrast media during angiography, and a more frequent use of preventive measures may be reasons for the decreasing incidence. Nevertheless, despite any effort

to reduce tubular toxicity, we detected a substantial increase in all biomarkers after the application of contrast media, which may be regarded as subclinical tubular injury. The clinical relevance of this tubular injury remains elusive.

The design of the present study aimed at an improvement of the current risk stratification with its subsequent preventive strategies. Therefore, the study design did not influence peri-procedural fluid administration, which was performed according to clinical standards with the physicians being blinded for urinary biomarkers. The study thereby inevitably comprised subjects with and without preventive plasma expansion, which, nevertheless constitutes a limitation. On the other hand, it has to be kept in mind that the AMACING study did not show any benefit of saline application at all. [28]

Current guidelines still recommend plasma expansion for subjects with clinically manifest CKD. [30] In those subjects without albuminuria > 30mg/g creatinine or an eGFR < 60 ml/min, urinary NGAL/creatinine ratios provide additional prognostic information. They were 2.6 fold higher in those subjects, who developed CI-AKI later on, noteworthy, without reaching statistical significance. Hence, a subclinical tubular injury is indeed associated with an increased vulnerability for contrast media-induced kidney injury. Thus, NGAL has a prognostic value both in presence and absence of clinically overt renal disease. Published predictive models for CI-AKI are not excelling and show conflicting results in external validation studies. [31, 32] An interesting common aspect is the high NPV, as seen in our study. We applied two scored based models and saw an increment in AUC by including urinary NGAL/creatinine ratios. Further trials, implementing renal biomarkers into common predictive models for CI-AKI, are needed.

There is certainly no reason to measure urinary NGAL concentrations in the overall population prior to application of contrast media. In individual critical clinical scenarios, however, the high negative predictive value of a NGAL/creatinine ratio < 56.4 μg/mg might indeed be helpful to decide for or against the use of contrast media.

The present study shows that NGAL but not KIM-1 or calprotectin provide prognostic information for the occurrence of AKI after contrast media administration. The high negative predictive value of a urinary NGAL/creatinine ratio < 56.4 μg/mg may be a clinically relevant information beyond the traditional risk stratification biomarkers eGFR and proteinuria.

## Supporting information

**S1 Table. Univariate and multivariable logistic regression model for CI-AKI.** Univariate logistic regression for CI-AKI using the biomarkers urinary NGAL-, KIM-1- and calprotectin/creatinine as predictors in the overall study population and in the subgroup of subjects without overt CKD. The area under the curve (AUC) represents the discriminatory accuracy of each biomarker, as well as the concordance statistic for the regression model using eGFR, NGAL-, KIM-1- and calprotectin/creatinine ratios as independent variables. Moreover, the predictive value of the models of Inohara et al. and Ghani et al. are presented with and without including NGAL/creatinine ratio into the model.
(DOCX)

## Acknowledgments

We thank Mrs. Voigt and Mrs. Dähnicke for their tireless devotion to this study.

## Author Contributions

**Conceptualization:** Felix S. Seibert, Nikolaos Pagonas, Timm H. Westhoff.

**Data curation:** Felix S. Seibert, Anja Heringhaus, Nikolaos Pagonas, Henrik Rudolf, Benjamin Rohn, Frederic Bauer, Nina Timmesfeld, Hans-Joachim Trappe.

**Formal analysis:** Felix S. Seibert, Henrik Rudolf, Nina Timmesfeld.

**Funding acquisition:** Timm H. Westhoff.

**Investigation:** Felix S. Seibert, Anja Heringhaus, Timm H. Westhoff.

**Methodology:** Felix S. Seibert, Anja Heringhaus, Nikolaos Pagonas, Henrik Rudolf, Nina Timmesfeld, Timm H. Westhoff.

**Project administration:** Felix S. Seibert, Benjamin Rohn, Frederic Bauer, Hans-Joachim Trappe, Nina Babel, Timm H. Westhoff.

**Software:** Felix S. Seibert, Henrik Rudolf, Nina Timmesfeld.

**Supervision:** Felix S. Seibert, Timm H. Westhoff.

**Visualization:** Felix S. Seibert.

**Writing – original draft:** Felix S. Seibert, Timm H. Westhoff.

**Writing – review & editing:** Felix S. Seibert, Henrik Rudolf, Nina Timmesfeld, Timm H. Westhoff.

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
