## [Decision Letter · Decision Letter 0]

6 Apr 2020

PONE-D-20-01191

Biomarkers in the prediction of contrast media induced nephropathy – the BITCOIN study

PLOS ONE

Dear Dr. Seibert,

Thank you for submitting your manuscript to PLOS ONE. After careful consideration, we feel that it has merit but does not fully meet PLOS ONE’s publication criteria as it currently stands. Therefore, we invite you to submit a revised version of the manuscript that addresses the points raised during the review process.

We would appreciate receiving your revised manuscript by May 21 2020 11:59PM. To enhance the reproducibility of your results, we recommend that if applicable you deposit your laboratory protocols in protocols.io, where a protocol can be assigned its own identifier (DOI) such that it can be cited independently in the future. For instructions see: http://journals.plos.org/plosone/s/submission-guidelines#loc-laboratory-protocols

We look forward to receiving your revised manuscript.

Kind regards,

Corstiaan den Uil

Academic Editor

PLOS ONE

3. Please ensure that you refer to Figure 3 in your text as, if accepted, production will need this reference to link the reader to the figure.

Reviewers' comments:

Reviewer's Responses to Questions

**Comments to the Author**

1. Is the manuscript technically sound, and do the data support the conclusions?

Reviewer #1: Yes

Reviewer #2: Partly

2. Has the statistical analysis been performed appropriately and rigorously? 

Reviewer #1: No

Reviewer #2: No

3. Have the authors made all data underlying the findings in their manuscript fully available?

Reviewer #1: Yes

Reviewer #2: Yes

4. Is the manuscript presented in an intelligible fashion and written in standard English?

Reviewer #1: Yes

Reviewer #2: Yes

5. Review Comments to the Author

Reviewer #1: The following points should be clarified:

- There is disconcordance between Result section and Tables. The sample has male or female predominance?

- What is the reason for urinary analysis of the markers in stead of blood sample?

- Please provide pre-procedural and post-precudural creatinine level.

- Regression analysis should be explaned with a table.

- Please provide the amount of contrast amount.

- Previous studies reported the usefullness of early post-precudural NGAL level. Do you think that, only preprocudural analysis would be enough?

- Classical AKI defined as 0.5 mg/dl increament in the Cr level. What is the reason for prefer 0.3 mg/dl instead of 0.5.

- The percentage of AKI is relatively lower although 0.3 mg/dl increament was proposed for AKI patients. What would be the AKI rate according to classical 0.5 mg/dl increament in serum Cr

- What would be the reason for the neutral effects of KIM-1 and calprotectin in the prediction of AKI

Reviewer #2: This is a nicely designed cohort study and used several novel biomarkers including KIM-1, NGAL and calprotectin to predict the risk of CI-AKI.

Some concerns need to be clarified.

1. There are numerous established predictive models for CI-AKI and some of them had pretty good predictive ability. The author may consider to provide the incremental value (i.e., the increase in AUROC, the integrated discrimination index, etc.) of each biomarker over the existing CI-AKI prediction models. Based on this, it will better illustrate the role of these new biomarkers in some clinical circumstances. In fact, the number of CI-AKI events are only 30, it will be limited to perform some rigorous statistical analyses. However, it is recommended for the authors to make some effort on this.

2. Did the author have post-coronary angiography urine or blood sample for the new biomarkers? I think it would be very interesting to show the proportion of patients with an elevated biomarker (compared with baseline) versus the proportion of those with an elevated creatinine.

Your definition of CI-AKI is based on the KDIGO AKi staging, however, it will be better to show that how many people might have a subclinical structural AKI (elevated biomarker, no elevated creatinine), and maybe another with only functional AKI (elevated creatinine, with normal biomarker).

Materials and methods

Statistical analysis

1. Since the statistical method of receiver operating characteristics (ROC) was used for discrimination or diagnosis, please replace “prediction” with “diagnosis” (or discrimination) and replace “predictive accuracy” with “diagnostic accuracy” throughout the manuscript.

2. It is unclear for the sentence “multiple binary logistic regression testing for CI-AKI in the presence of eGFR, NGAL-, KIM-1-, and calprotectin/creatinine ratios”. Do the authors introduced these four parameters into one multivariable model? If so, how was the correlation (is there a collinearity problem?) among the three markers? The result of multivariable logistic regression model should be provided, maybe in the supplement file.

3. Please replace “multiple binary logistic regression testing” with “multivariable logistic regression model”

4. Please clarify which method of significance of AUC was used, such as the DeLong’s method.

Results

1. Provide the 95% confidence interval of AUC throughout the text.

2. Please directly compare the AUCs among the three markers, the DeLong’s test may be appropriate.

Table 1

1. The structure of Table 1 is complicated and poorly presented. Only five columns are needed: the variable name, the data of total patients, the data of CI-AKI patients, the data of non CI-AKI patients and the p value.

Table 2

1. Which test (the column with p value) was performed, is that Mann-Whitney U-test? If so, where is the result of multiple binary logistic regression?

2. Please provide 3 digits value of P value

3. The PPV/NPV is calculated based on a given prevalence, please clarify how the prevalence is determined. The prevalence value should be based on a population data or at least a large-scale representative study.

Figure and Figure legends

1. Please give the exact p value with three digits in Figure 1 and Figure 3 for each marker, regardless of the group comparison is significant or not.

2. Provide the 95% confidence interval of AUC in Figure 2 or its figure legend

6. PLOS authors have the option to publish the peer review history of their article (what does this mean?). If published, this will include your full peer review and any attached files.

Reviewer #1: No

Reviewer #2: No

---

## [Author Response · Author response to Decision Letter 0]

26 May 2020

The Response to the Reviewers is also uploaded in a separate file.

--

Reviewer #1: The following points should be clarified:

- There is disconcordance between Result section and Tables. The sample has male or female predominance?

REPLY: Thank you for drawing our attention to this flaw. Changes have been made in the revised version of the manuscript. There is a predominance of male subjects.

- What is the reason for urinary analysis of the markers instead of blood sample?

REPLY: Analysing renal biomarkers in the urine is a non-invasive and easy method to perform. The rationale to investigate the urine is to focus on its specific organ: the kidneys. Hence, KIM-1 is highly upregulated in proximal tubular cells following kidney injury and is shed into the lumen. NGAL and calprotectin are molecules of the immune system, which can be detected in the blood for inflammation of any kind in the human body. For instance, patients with SIRS show high NGAL values in plasma, while urinary NGAL seems less affected by the systemic inflammation.[1] In other words, the diagnostic accuracy of a renal biomarker is sought to be higher in the urine. There are large studies demonstrating, that urinary NGAL shows a higher diagnostic performance (e.g. sensitivity, specificity, AUC) compared to plasma in the setting of AKI.[2] Studies about the investigation of calprotectin in AKI are rare, and have never been performed in the specific setting of CI-AKI before.[3-6] On the one hand, systemic inflammation enhances calprotectin production, which is a danger associated molecule pattern protein. On the other hand, it is a measure of local inflammatory activity, that seems to be unaffected by a variety of conditions that result in an elevation of systemic inflammation.[7, 8] In summary, it is reasonable for all the investigated biomarkers to be analysed in the urine. We clarified this issue in the modified version of the manuscript.

- Please provide pre-procedural and post-procedural creatinine level.

REPLY: Pre- and postprocedural creatinine were provided (Table 1) for the overall study population and the subpopulation without overt CKD in the new version of the manuscript as requested.

- Regression analysis should be explained with a table.

REPLY: A detailed table about the results of the binary logistic regression has been added to the manuscript (Supplements Table 1).

- Please provide the amount of contrast amount.

REPLY: The median amount of the contrast media has been implemented in the Results, not only for the overall population, but for the subpopulation without overt CKD as well.

- Previous studies reported the usefulness of early post-procedural NGAL level. Do you think that, only preprocedural analysis would be enough?

REPLY: We focussed on preprocedural analysis, since the biomarker examination intended the opportunity to identify subjects, who might benefit from periprocedural preventive measures (prognostic value). Post-procedural investigations of the biomarkers are indeed useful to detect subclinical AKI. We decided for the robust primary endpoint of clinically manifest AKI. Neverthess, we additionally provide postprocedural biomarker concentrations in the revised version of the manuscript. 

- Classical AKI defined as 0.5 mg/dl increment in the Cr level. What is the reason for prefer 0.3 mg/dl instead of 0.5.

REPLY: The definition of acute kidney injury is beyond any doubt an important aspect of the study. The reviewer probably refers to the definition acute renal failure (ARF) according to the Risk/Injury/Failure/Loss/End-stage (RIFLE) criteria. The sensitivity and accuracy of the RIFLE compared to AKIN criteria for acute kidney injury (AKI) remains uncertain.[9] In our study, we decided to follow the international recognized KDIGO 2012 guidelines, which use AKIN criteria for the definition and the classification of AKI.[10] Moreover, RIFLE criteria are defined as changes within 7 days, while the AKIN criteria suggest using 48 hours, which is a better fit to our study design.

- The percentage of AKI is relatively lower although 0.3 mg/dl increment was proposed for AKI patients. What would be the AKI rate according to classical 0.5 mg/dl increment in serum Cr

REPLY: Using a 0.5 mg/dl increment in serum creatinine as a definition, we would face a lower incidence of AKI (n=8). We described this aspect in the modified version of the manuscript.. As described in Material and Methods and elucidated in the previous question, we primarily followed the international recognized guidelines KDIGO 2012 and defined an AKI according to the AKIN criteria.[10]

- What would be the reason for the neutral effects of KIM-1 and calprotectin in the prediction of AKI.

REPLY: This is indeed an interesting fact, for which there is no clear explanation yet. KIM-1 is a tubular marker like NGAL, although it is expressed at a different location in the nephron. We sought to reflect proximal (KIM-1) and distal (NGAL) tubular damage. It can be speculated, that the difference in-between the renal tubular biomarkers is the result of contrast media toxicity at different levels. Current data suggests, that the deleterious effect of contrast media is more pronounced at the distal tubule, partly due to oxidative stress, impaired tubuloglomerular feedback and increasing viscosity along the tubule.[11, 12] Noteworthy, however, there is a non-significant trend for the prediction of CI-AKI by KIM-1. Calprotectin mirrors inflammation, rather than a pre-existing tubular damage. A negative result is plausible.

Reviewer #2: This is a nicely designed cohort study and used several novel biomarkers including KIM-1, NGAL and calprotectin to predict the risk of CI-AKI.

Some concerns need to be clarified.

1. There are numerous established predictive models for CI-AKI and some of them had pretty good predictive ability. The author may consider to provide the incremental value (i.e., the increase in AUROC, the integrated discrimination index, etc.) of each biomarker over the existing CI-AKI prediction models. Based on this, it will better illustrate the role of these new biomarkers in some clinical circumstances. In fact, the number of CI-AKI events are only 30, it will be limited to perform some rigorous statistical analyses. However, it is recommended for the authors to make some effort on this.

REPLY: We thank the reviewer for his thoughtful remark and suggestion. The meta-analysis of Allen et al. describes 30 models predicting CI-AKI, from which 9 studies only used preprocedural variables, analogously to our study design.[13] Noteworthy, including postprocedural information (remaining 21 studies) did not significantly augment the discriminatory power. There is a considerable amount of heterogeneity amongst the trials, and definition of CI-AKI is not homogenous either. The AUC of NGAL/creatinine (0.68 (95% CI 0.60 - 0.81]) in our work is lower than the c-statistic of the 9 trials, mentioned above, ranging from 0.71 (95% CI 0.71 – 0.72) to 0.88 (95% CI 0.85 – 0.91). However, there was an insufficient external validation for these models. Interestingly, there is a considerable discriminating ability to identify “true-negative” cases (high NPV), as we have seen in our study. The most recent study about CI-AKI prediction model is only a few days old and applies 17 risk prediction models for CI-AKI.[14] We chose 2 different prediction models with preprocedural variables only, fitting the available data. NGAL showed an increase in AUC for both models, which was significant by the DeLong comparison method for the Ghani et al. model. The manuscript has been changed accordingly.

2. Did the author have post-coronary angiography urine or blood sample for the new biomarkers? I think it would be very interesting to show the proportion of patients with an elevated biomarker (compared with baseline) versus the proportion of those with an elevated creatinine.

Your definition of CI-AKI is based on the KDIGO AKI staging, however, it will be better to show that how many people might have a subclinical structural AKI (elevated biomarker, no elevated creatinine), and maybe another with only functional AKI (elevated creatinine, with normal biomarker).

REPLY: The reviewer addresses the interesting aspect of subclinical renal injury post contrast media and we, therefore, have analysed postprocedural urine of this study population. One possibility to assess the presence of renal damage is to follow the cardiac surgery-associated NGAL score (CSA-NGAL score), where NGAL levels >100 ng/mL suggest a renal tubular damage, however there is no data about the NGAL/ creatinine ratio.[15] We therefore defined an elevation of NGAL/creatinine by 100% (2 fold) as subclinical tubular injury, and detected 131 subclinical kidney injuries. We performed the same procedure for KIM-1. Since calprotectin is no marker or tubular damage itself, we refrained from this analysis.

Materials and methods

Statistical analysis

1. Since the statistical method of receiver operating characteristics (ROC) was used for discrimination or diagnosis, please replace “prediction” with “diagnosis” (or discrimination) and replace “predictive accuracy” with “diagnostic accuracy” throughout the manuscript.

REPLY: We thank the reviewer for his remark. ROC analysis is indeed traditionally used to describe diagnostic accuracy. It is possible, however, to test prognostic accuracy as well.[16, 17] The present study’s goal was not to diagnose a CI-AKI, but to predict a potential loss of renal function due to contrast media application. In other words, we did not analyse renal biomarkers after the application of contrast media, but before the procedure was done. We offer changing the words according to the suggestion of the reviewer, but are afraid that this might be misleading for some readers and would therefore prefer to keep the word “prediction”.

2. It is unclear for the sentence “multiple binary logistic regression testing for CI-AKI in the presence of eGFR, NGAL-, KIM-1-, and calprotectin/creatinine ratios”. Do the authors introduced these four parameters into one multivariable model? If so, how was the correlation (is there a collinearity problem?) among the three markers? The result of multivariable logistic regression model should be provided, maybe in the supplement file.

REPLY: All four parameters have been integrated into a single multivariable model. The Pearson correlation in-between the parameters is low (<0.5 each, e.g. NGAL/creatinine & eGFR 0.104, KIM-1/creatinine & NGAL/creatinine 0.262, calprotectin/creatinine & KIM-1/creatinine 0.142). We added this information to the manuscript.

3. Please replace “multiple binary logistic regression testing” with “multivariable logistic regression model”

REPLY: We thank the reviewer for this remark. The manuscript has been changed accordingly.

4. Please clarify which method of significance of AUC was used, such as the DeLong’s method.

REPLY: We used the DeLong method and mentioned this in the revised version of the manuscript.[18]

Results

1. Provide the 95% confidence interval of AUC throughout the text.

REPLY: Changes have been made throughout the manuscript according to the reviewers remarks.

2. Please directly compare the AUCs among the three markers, the DeLong’s test may be appropriate.

REPLY: The comparison of the AUCs by ROC were calculated according to the DeLong et al.[18] Only the pairwise comparison of ROC curves from NGAL/creatinine and calprotectin/creatinine became significant with p=0.009. The other pairings (KIM-1/creatinine & NGAL/creatinine and KIM-1/creatinine & calprotectin/creatinine) did not show any relevant differences in AUC (p=0.181 and p=0.577). The manuscript has been changed accordingly.

Table 1

1. The structure of Table 1 is complicated and poorly presented. Only five columns are needed: the variable name, the data of total patients, the data of CI-AKI patients, the data of non CI-AKI patients and the p value.

REPLY: Table 1 has been redesigned according to the reviewers remarks.

Table 2

1. Which test (the column with p value) was performed, is that Mann-Whitney U-test? If so, where is the result of multiple binary logistic regression?

REPLY: As described in Material and Methods, comparison of continuous parameters in subjects with and without CI-AKI was performed by Mann Whitney U-test. The results of the regression model is presented in the supplement section.

2. Please provide 3 digits value of P value

REPLY: Changes have been made throughout the manuscript according to the reviewers remarks.

3. The PPV/NPV is calculated based on a given prevalence, please clarify how the prevalence is determined. The prevalence value should be based on a population data or at least a large-scale representative study.

REPLY: We have been made to the manuscript. PPV and NPV calculations are based on the sensitivity and specificity of the analysed biomarker within the population. We chose the cut-off value using the Youden index.

Figure and Figure legends

1. Please give the exact p value with three digits in Figure 1 and Figure 3 for each marker, regardless of the group comparison is significant or not.

REPLY: Changes have been made to the two figures according to the reviewer’s remarks.

2. Provide the 95% confidence interval of AUC in Figure 2 or its figure legend

REPLY: Changes have been made to Figure 2 according to the reviewer’s remarks.

 

REFERENCES

1. Martensson J, Bell M, Oldner A, Xu S, Venge P, Martling CR. Neutrophil gelatinase-associated lipocalin in adult septic patients with and without acute kidney injury. Intensive Care Med. 2010;36(8):1333-40. Epub 2010/04/17. doi: 10.1007/s00134-010-1887-4. PubMed PMID: 20397003.

2. Zhang A, Cai Y, Wang PF, Qu JN, Luo ZC, Chen XD, et al. Diagnosis and prognosis of neutrophil gelatinase-associated lipocalin for acute kidney injury with sepsis: a systematic review and meta-analysis. Crit Care. 2016;20:41. Epub 2016/02/18. doi: 10.1186/s13054-016-1212-x. PubMed PMID: 26880194; PubMed Central PMCID: PMCPMC4754917.

3. Seibert FS, Pagonas N, Arndt R, Heller F, Dragun D, Persson P, et al. Calprotectin and neutrophil gelatinase-associated lipocalin in the differentiation of pre-renal and intrinsic acute kidney injury. Acta Physiol (Oxf). 2013;207(4):700-8. Epub 2013/01/23. doi: 10.1111/apha.12064. PubMed PMID: 23336369.

4. Heller F, Frischmann S, Grunbaum M, Zidek W, Westhoff TH. Urinary calprotectin and the distinction between prerenal and intrinsic acute kidney injury. Clin J Am Soc Nephrol. 2011;6(10):2347-55. Epub 2011/09/03. doi: 10.2215/CJN.02490311. PubMed PMID: 21885792; PubMed Central PMCID: PMCPMC3359561.

5. Westhoff JH, Seibert FS, Waldherr S, Bauer F, Tonshoff B, Fichtner A, et al. Urinary calprotectin, kidney injury molecule-1, and neutrophil gelatinase-associated lipocalin for the prediction of adverse outcome in pediatric acute kidney injury. Eur J Pediatr. 2017;176(6):745-55. Epub 2017/04/15. doi: 10.1007/s00431-017-2907-y. PubMed PMID: 28409285.

6. Seibert FS, Rosenberger C, Mathia S, Arndt R, Arns W, Andrea H, et al. Urinary Calprotectin Differentiates Between Prerenal and Intrinsic Acute Renal Allograft Failure. Transplantation. 2017;101(2):387-94. Epub 2016/02/24. doi: 10.1097/TP.0000000000001124. PubMed PMID: 26901081.

7. Gisbert JP, McNicholl AG. Questions and answers on the role of faecal calprotectin as a biological marker in inflammatory bowel disease. Dig Liver Dis. 2009;41(1):56-66. Epub 2008/07/08. doi: 10.1016/j.dld.2008.05.008. PubMed PMID: 18602356.

8. Tibble JA, Sigthorsson G, Foster R, Forgacs I, Bjarnason I. Use of surrogate markers of inflammation and Rome criteria to distinguish organic from nonorganic intestinal disease. Gastroenterology. 2002;123(2):450-60. Epub 2002/07/30. doi: 10.1053/gast.2002.34755. PubMed PMID: 12145798.

9. Xiong J, Tang X, Hu Z, Nie L, Wang Y, Zhao J. The RIFLE versus AKIN classification for incidence and mortality of acute kidney injury in critical ill patients: A meta-analysis. Sci Rep. 2015;5:17917. Epub 2015/12/08. doi: 10.1038/srep17917. PubMed PMID: 26639440; PubMed Central PMCID: PMCPMC4671151.

10. Mehta RL, Kellum JA, Shah SV, Molitoris BA, Ronco C, Warnock DG, et al. Acute Kidney Injury Network: report of an initiative to improve outcomes in acute kidney injury. Crit Care. 2007;11(2):R31. Epub 2007/03/03. doi: 10.1186/cc5713. PubMed PMID: 17331245; PubMed Central PMCID: PMCPMC2206446.

11. Liu ZZ, Schmerbach K, Lu Y, Perlewitz A, Nikitina T, Cantow K, et al. Iodinated contrast media cause direct tubular cell damage, leading to oxidative stress, low nitric oxide, and impairment of tubuloglomerular feedback. Am J Physiol Renal Physiol. 2014;306(8):F864-72. Epub 2014/01/17. doi: 10.1152/ajprenal.00302.2013. PubMed PMID: 24431205; PubMed Central PMCID: PMCPMC4422341.

12. Persson PB, Hansell P, Liss P. Pathophysiology of contrast medium-induced nephropathy. Kidney Int. 2005;68(1):14-22. Epub 2005/06/16. doi: 10.1111/j.1523-1755.2005.00377.x. PubMed PMID: 15954892.

13. Allen DW, Ma B, Leung KC, Graham MM, Pannu N, Traboulsi M, et al. Risk Prediction Models for Contrast-Induced Acute Kidney Injury Accompanying Cardiac Catheterization: Systematic Review and Meta-analysis. Can J Cardiol. 2017;33(6):724-36. Epub 2017/05/27. doi: 10.1016/j.cjca.2017.01.018. PubMed PMID: 28545621.

14. Serif L, Chalikias G, Didagelos M, Stakos D, Kikas P, Thomaidis A, et al. Application of 17 Contrast-Induced Acute Kidney Injury Risk Prediction Models. Cardiorenal Med. 2020:1-13. Epub 2020/04/15. doi: 10.1159/000506379. PubMed PMID: 32289786.

15. de Geus HR, Ronco C, Haase M, Jacob L, Lewington A, Vincent JL. The cardiac surgery-associated neutrophil gelatinase-associated lipocalin (CSA-NGAL) score: A potential tool to monitor acute tubular damage. J Thorac Cardiovasc Surg. 2016;151(6):1476-81. Epub 2016/03/10. doi: 10.1016/j.jtcvs.2016.01.037. PubMed PMID: 26952930.

16. Hajian-Tilaki K. Receiver Operating Characteristic (ROC) Curve Analysis for Medical Diagnostic Test Evaluation. Caspian J Intern Med. 2013;4(2):627-35. Epub 2013/09/07. PubMed PMID: 24009950; PubMed Central PMCID: PMCPMC3755824.

17. Zou KH, O'Malley AJ, Mauri L. Receiver-operating characteristic analysis for evaluating diagnostic tests and predictive models. Circulation. 2007;115(5):654-7. Epub 2007/02/07. doi: 10.1161/CIRCULATIONAHA.105.594929. PubMed PMID: 17283280.

18. DeLong ER, DeLong DM, Clarke-Pearson DL. Comparing the areas under two or more correlated receiver operating characteristic curves: a nonparametric approach. Biometrics. 1988;44(3):837-45. Epub 1988/09/01. PubMed PMID: 3203132.

---

## [Editor Report · Decision Letter 1]

5 Jun 2020

Biomarkers in the prediction of contrast media induced nephropathy – the BITCOIN study

PONE-D-20-01191R1

Dear Dr. Seibert,

We’re pleased to inform you that your manuscript has been judged scientifically suitable for publication and will be formally accepted for publication once it meets all outstanding technical requirements.

Kind regards,

Corstiaan den Uil

Academic Editor

PLOS ONE
---

## [Editor Report · Acceptance letter]

22 Jun 2020

PONE-D-20-01191R1 

Biomarkers in the prediction of contrast media induced nephropathy – the BITCOIN study 

Dear Dr. Seibert:

I'm pleased to inform you that your manuscript has been deemed suitable for publication in PLOS ONE. Congratulations! Your manuscript is now with our production department. 

Kind regards, 

on behalf of

Dr. Corstiaan den Uil 

Academic Editor

PLOS ONE